# MicroRNAs Expression Profile in MN1-Altered Astroblastoma

**DOI:** 10.3390/biomedicines13010112

**Published:** 2025-01-06

**Authors:** Francesca Gianno, Evelina Miele, Claudia Sabato, Elisabetta Ferretti, Simone Minasi, Francesca Romana Buttarelli, Debora Salerno, Natalia Pediconi, Giuseppe Rubens Pascucci, Francesca Guerrieri, Andrea Ciolfi, Simone Pizzi, Maura Massimino, Veronica Biassoni, Elisabetta Schiavello, Marco Gessi, Sofia Asioli, Angela Mastronuzzi, Antonio d’Amati, Giuseppina Catanzaro, Elisabetta Viscardi, David Capper, Felice Giangaspero, Manila Antonelli

**Affiliations:** 1Department of Radiological, Oncological and Anatomo-Pathological Sciences, University Sapienza of Rome, 00161 Rome, Italy; simone.minasi@uniroma1.it (S.M.); manila.antonelli@uniroma1.it (M.A.); 2IRCCS Neuromed, Via Atinense, 18, 86077 Pozzilli, Italy; 3Hematology/Oncology and Stem Cell Transplantation, Bambino Gesù Children’s Hospital, IRCCS, 00165 Rome, Italy; 4Department of Experimental Medicine, Sapienza University of Rome, 00161 Rome, Italy; claudia.sabato@crob.it (C.S.);; 5Laboratory of Preclinical and Translational Research, Centro di Riferimento Oncologico della Basilicata, IRCCS CROB, 85028 Rionero in Vulture, Italy; 6Department of Molecular Medicine, Sapienza University of Rome, 00161 Rome, Italy; 7Center for Life Nano- & Neuro-Science, Fondazione Istituto Italiano di Tecnologia (IIT), 16161 Rome, Italy; 8Research Unit of Clinical Immunology and Vaccinology, Academic Department of Pediatrics, Bambino Gesù Children’s Hospital, 00165 Rome, Italy; 9Department of Systems Medicine, University of Rome Tor Vergata, 00133 Rome, Italy; 10UMR INSERM U1052/CNRS 5286, Cancer Research Center of Lyon, 69008 Lyon, France; 11Molecular Genetics and Functional Genomics, Bambino Gesù Children’s Hospital, IRCCS, 00165 Rome, Italy; 12Institute of Biosciences and Bioresources, National Research Council, 06128 Perugia, Italy; 13Pediatric Unit, Fondazione IRCCS Istituto Nazionale Tumori, 20133 Milano, Italyveronica.biassoni@istitutotumori.mi.it (V.B.);; 14Department of Woman and Child Health Sciences and Public Health, Fondazione Policlinico Universitario A. Gemelli IRCCS, 00168 Rome, Italy; 15IRCCS Istituto di Scienze Neurologiche di Bologna, 40139 Bologna, Italy; sofia.asioli3@unibo.it; 16Departmemt of Biomedical e Neurmotor Science, Alma Mater University of Bologna, 40126 Bologna, Italy; 17Hematology/Oncology, Cell Therapy, Gene Therapies and Hemopoietic Transplant, 586724 Bambino Gesù Children’s Hospital, 00165 Rome, Italy; 18Department of Basic Medical Sciences, Neuroscience, and Sensory Organs, University of Bari School of Medicine, 70121 Bari, Italy; 19Department of Medicine and Surgery, LUM University, 70010 Bari, Italy; 20Department of Life Science, Health, and Health Professions, Link Campus University, 00165 Rome, Italy; g.catanzaro@unilink.it; 21Pediatric Oncology Unit, Padova University, 35122 Padova, Italy; 22Charité–Universitätsmedizin Berlin, Corporate Member of Freie Universität Berlin and Humboldt, 10117 Berlin, Germany; 23German Cancer Consortium (DKTK), Partner Site Berlin, German Cancer Research Center (DKFZ), 69120 Heidelberg, Germany

**Keywords:** astroblastoma, microRNA, miRNA expression, biological processes, epigenetic, methylation, MN1 rearrangement

## Abstract

Background/Objectives: Astroblastoma is a rare glial neoplasm more frequent in young female patients, with unclear clinical behaviors and outcomes. The diagnostic molecular alteration is a rearrangement of the Meningioma 1 (*MN1*) gene. MicroRNAs (miRNAs) are important gene expression regulators with strong implications in biological processes. Here, we investigated microRNA expression, regulation, and biological processes correlated to target genes of deregulated miRNAs in MN1-altered astroblastoma. Methods: A cohort of 14 tumor samples, histologically classified as astroblastoma, was retrospectively collected and analyzed through their DNA methylation profiles. MiRNA expression profiles were then detected on *MN1*-altered astroblastomas (*n* = 8) and normal brain controls (*n* = 2) by Nanostring technology and validated by RT-qPCR; then, the expression of deregulated miRNAs was correlated with clinical-pathological characteristics. Subsequently, the methylation status of promoters of deregulated miRNAs was investigated through a methylation profiling microarray. Finally, bioinformatics analysis was conducted to explore the biological processes (BPs) and target genes of differentially expressed miRNAs. Results: Eight *MN*-altered astroblastoma were identified. Thirty-nine miRNAs were deregulated in tumor samples compared to normal brain tissue. Downregulated microRNAs exhibited an association with an increased risk of recurrence. The promoter methylation status was investigated in 32/39 miRNAs: 14/32 were epigenetically deregulated. None of them were genetically regulated. Conclusions: *MN1*-altered astroblastomas have an miRNA expression signature that identifies specific BPs and pathways. Our findings suggested that the involved pathways could be associated with clinical and pathological characteristics of MN1-altered astroblastomas. Also, the biology of this rare tumor could have potential implications on prognostic markers and therapy.

## 1. Introduction

Astroblastoma is a rare glial neoplasm with an incidence between 0.45% and 2.8% of primary brain tumors [1], a median age of 15 years (age range: 3 months–40 years) [2,3,4,5] and a female predominance. It occurs predominantly in the cerebral hemispheres as a well-demarcated mass [6].

The oncogenic driver event involves the meningioma (disrupted in balanced translocation) 1 (*MN1*) gene, a transcriptional co-regulator localized at chromosome 22q12.1. In astroblastoma, the *MN1* gene is usually rearranged in-frame with the *BEND2* gene at chromosome Xp22.13, and less frequently with *CXXC5* [7,8]. The histologic characteristic of MN1-altered astroblastoma is the presence of “astroblastic” pseudorosettes [3,7,9,10,11]. However, the histological and immunohistochemical features are not pathognomonic of this tumor [3,10,12]. Outcome data are limited, but it has been shown that high-grade histology is associated with recurrence and tumor progression [13]. More than 70% of these tumors show a distinct DNA methylation profile [3,7,10,11] and recurrent chromosomal copy number changes, including monosomy chromosome 16 and partial losses of 22q and X.

MicroRNAs are small non-coding RNAs containing 2 nucleotides that modulate RNA silencing and post-transcriptional regulation of gene expression [14]. MiRNAs play a crucial role as regulators of gene expression by binding to complementary sequences on target messenger RNAs (mRNAs), resulting in mRNA degradation or translational repression. In tumor development, miRNAs can act as oncogenes or tumor suppressors, depending on their target genes. Dysregulation of miRNA expression has been implicated in multiple mechanisms, including alterations in cell proliferation, apoptosis, invasion, and angiogenesis [15]. MiRNAs do not act in isolation, but are part of intricate regulatory networks, interacting with other non-coding RNAs, transcription factors, and signaling pathways. Dysregulation of these networks often underpins tumor initiation and progression [16]. MiRNAs are also emerging as promising targets for cancer therapy: miRNA mimics restore tumor suppressor miRNAs, while antagomiRs inhibit oncomiRs to suppress tumor growth [17,18].

High-throughput technologies provide instruments for studying the global miRNA profile, identifying tissue origin of human cancers and identifying the association with clinical outcomes [19,20,21,22]. Aberrant expression of miRNAs in cancer can be caused by several mechanisms [23], including genetic abnormalities (e.g., deletion, amplification, translocation of miRNAs loci) [24,25,26], epigenetics (e.g., DNA methylation of regulatory regions) [27,28,29,30], transcriptional regulation by transcription factors (TFs) [31,32,33,34] and altered miRNAs processing steps (e.g., mutation in *DICER*, *DROSHA*) [35].

Previous studies on miRNA expression profiles in other gliomas demonstrated that deregulated miRNAs function as an oncogene in glioblastomas by blocking the expression of key apoptosis-enabling genes [36]. Also, their role in tumor biology has highlighted the significance of specific miRNA signatures in distinguishing glioma subtypes and their potential as prognostic markers for the risk of disease progression. In diffuse intrinsic pontine glioma (DIPG), a miRNA signature identifies patients who are likely to respond poorly to radiotherapy. This enables clinicians to avoid unnecessary additional courses of treatment or re-irradiation at relapse, which has become a common practice for patients with recurrent DIPG [37]. Also, investigation in low-grade gliomas demonstrated the involvement of miRNAs in pathogenesis, identifying key miRNAs associated with tumor progression and patient survival [38]. Further research revealed the interplay between miRNA dysregulation and epigenetic alterations in gliomas, emphasizing their contribution to the molecular heterogeneity of these tumors [39].

We explored the expression pattern of miRNAs in a cohort of *MN-1* altered astroblastoma to identify differentially expressed miRNAs versus normal tissues, as well as to shed light on the mechanisms underlying the aberrant expression. Biological processes and target genes associated with deregulated miRNAs were also investigated.

## 2. Materials and Methods

### 2.1. Patients and Samples

Formalin-fixed paraffin-embedded (FFPE) tissue samples were collected retrospectively (from 2003 to 2015) from the Italian National Program of Centralization of Paediatric Brain Tumour, contributed to by Sapienza University of Rome (Rome, Italy), IRCCS C. Besta (Milan, Italy), Ospedale di Circolo Fondazione Macchi di Varese (Varese, Italy), IRCCS Neuromed (Pozzilli, Italy), Fondazione Policlinico Universitario “A. Gemelli” IRCCS (Rome, Italy), Istituto Giannina Gaslini (Genua, Italy), Trento Hospital (Trento, Italy) and Bellaria Hospital (Bologna, Italy). The samples included 14 cases histologically defined as astroblastoma. Clinical and pathologic features are summarized in Table 1. The study received full approval from the local ethics committee of Policlinico Umberto I of Rome (approval code: Prot 0102463; approval date: 24 June 2020). The subjects or their parents provided written informed consent to the use of their biological material and data for research purposes.

### 2.2. Histology and Immunohistochemistry

FFPE 3-mm-thick sections from tissue samples were cut and stained with haematoxylin and eosin (HE), with a pre-step of deparaffination with xylol. Histology was reviewed independently by two pathologists (FG, MA), and an immunohistochemical panel including Glial Fibrillary Acid Protein (GFAP), Oligodendrocyte transcription factor 2 (OLIG2), Epithelial Membrane Antigen (EMA), Estrogenic Receptor (ER) and Ki67, was performed.

Immunohistochemistry (IHC) was performed by a Leica Bond RXm™ automated staining processor (Leica Biosystems, Buffalo Grove, IL, USA). Tissue sections were cut at 5 μm, dried at 70 °C for 30 min and then dewaxed. Antigen retrieval was performed in the Bond Rx system with Epitope Retrieval Solution 1 (pH 6) for 30 min. Sections were incubated for 30 min with GFAP (Novocastra-Leica Biosystems clone GA5, mouse monoclonal, 1:400 dilution), OLIG2 (18953-S IBL America, Minneapolis, MN, USA, rabbit polyclonal, 1:400 dilution), EMA (BOND-Leica Biosystem clone GP1.4, monoclonal antibody, ready to use) and ER (Leica Biosystem clone 6F11-mouse monoclonal antibody, ready to use).

### 2.3. DNA Methylation Profiling

The DNA methylation profile was performed according to protocols approved by the institutional review board with written consent obtained from the patients and patient’s parents. Tumor areas with the highest tumoral cell content (≥70%) were selected for DNA extraction.

DNA was extracted from FFPE blocks after manual microdissection using the QIAamp DNA FFPE Tissue Kit protocol (Cat# 56404, QIAGEN). Concentrations were measured using a dsDNA (BR) assay kit by a Qubit fluorometer (Cat# Q32850, Life Technologies, Carlsbad, CA, USA). In detail, 250 ng of DNA was used as input material for the methylation profile. Bisulphite-converted DNA was amplified, fragmented, and hybridized using the Infinium Human Methylation 450K Bead Chip according to the manufacturer’s instructions (Datasheet_humanmethylation450), on the Illumina iScan Platform (San Diego, CA, USA). We imported raw data IDAT files for subsequent analysis by means of R package ChAMP, using default quality control and normalization steps [40]. We checked batch effects by means of the singular value decomposition method, to exclude any significant relevance of technical factors on methylation differences [34]. Generated methylation data were compared with the Heidelberg brain tumor classifier (v11b4) to assign a subgroup score for the tumor compared to 91 different brain tumor entities [41].

### 2.4. MiRNAs Promoter Methylation Status Analysis

The Infinium HumanMethylation450 BeadChip assay provides genome-wide coverage, featuring comprehensive gene region and CpG islands. Coverage is targeted across gene regions with sites in the promoter region, 5′UTR, first exon, gene body, and 3′UTR to provide a wide and complete vision of methylation status. This multiple-site approach was extended to include multiple additional content categories, like miRNAs promoter regions. In our cohort, we retrieved known miRNAs promoter regions and analyzed their methylation status in all MN1-altered astroblastomas and 6 normal hemispheric cortex samples extrapolated from the published dataset [41]. All CpG islands upstream of the TSS up to a maximum of 1500 bases were considered. The methylation levels were quantified by the beta-value ranging from 0 to 1 (values < 0.2 designated a hypomethylated status; values > 0,8 designated a hypermethylated status).

### 2.5. Copy Number Variation Analysis

From the Illumina Infinium Human Methylation array data, it is also possible to extrapolate a Copy Number Variation (CNV) plot from un-normalized signal intensities. Signal intensities were extracted for each sample using GenomeStudio software version 2.0.3 and normalized [42]. Thresholds for the identification of single-copy CNV were derived from the difference between normal reference male and female DNA specimens involving a single copy change in the X chromosome. This method for identifying CNV from the Infinium methylation arrays is incorporated using the ChAMP Bioconductor package (Bioconductor. http://www.bioconductor.org/), an Infinium Human Methylation 450K and EPIC array integrated analysis pipeline, leading to the detection of CNV [40]. An Integrative Genomic Viewer (IGV) was used to visualize structural rearrangements and map genes onto regions of interest (microRNA mapping regions).

### 2.6. MiRNAs Profiling

The multiplexed NanoString nCounter miRNAs expression assay (NanoString Technologies, Seattle, WA, USA) was used to profile more than 700 human miRNAs. The assay was performed according to the manufacturer’s protocol. Briefly, total RNA was extracted by an RNeasy FFPE kit (Cat# 73504, QIAGEN, Hilden, Germany) according to the manufacturer’s protocol from the paraffin embedded tissue. A NanoDrop spectrophotometer (ND2000; Wilmington, DE, USA) was used for quantification of the extracted RNA. To perform miRNAs profile, 100 ng of total RNA was used as input material. A specific DNA tag was ligated onto the 3′end of each mature miRNA, providing an exclusive identification for each miRNA species in the sample. The tagging was performed in a multiplexed ligation reaction, utilizing reverse complementary bridge oligonucleotides, to dispose the ligation of each miRNA to its designated tag. All hybridization reactions were incubated at 65 °C for 18 h. Excess tags were then removed, and the resulting material was hybridized with a panel of fluorescently labelled, bar-coded reporter probes specific to the miRNAs of interest.

### 2.7. MiRNAs Data Analysis

Abundances of miRNAs were quantified with the nCounter Prep Station and Digital Analyzer (NanoString) by counting individual fluorescent barcodes and quantifying target miRNAs molecules present in each sample. For each assay, a high-density scan (555 fields of view) was performed. Each sample was normalized to the geometric mean of the top 100 highly expressed miRNAs.

MiRNA counts were transformed to log_2_ scale, and subsequently, differential analyses between 8 MN1-altered astroblastoma and 2 controls (normal brain tissue) were performed using the Wilcoxon test.

Significant differentially expressed miRNAs were identified as either up- or down-regulated, with fold change values greater than 2 and less than −2, respectively, with *p*-values  <  0.05. Hierarchical clustering of deregulated miRNAs based on average linkage and Euclidian distance was performed using an online tool (http://www.heatmapper.ca/expression/ (accessed on 1 January 2021))

### 2.8. Gene Ontology and Pathway Enrichment Analysis

Functional enrichment of biological processes (BPs) Gene Ontology (GO) for differentially expressed (DE) miRNAs was performed using the R open-source software (version 3.6.2) by Cluster Profiler package. GO terms were based on *p*-value < 0.05, and the adjustment method used was False Discovery Rate (FDR). Only validated targets (from MiRTarBase) for their BPs were examined.

KEGG pathway analysis of deregulated miRNAs was performed by DIANA-miRPath web server v.3. Only experimentally validated miRNAs interactions derived from DIANA-TarBase v7.0 were considered. DIANA-miRPath v3.0 applies the Fisher’s Exact Test, EASE score and FDR methodologies [43].

### 2.9. Real-Time Quantitative Polymerase Chain Reaction (RT-qPCR)

To validate top significant candidates obtained by the NanoString nCounter miRNAs assay, the RT-qPCR was performed. Validation of the expression of selected miRNAs markers was performed using TaqMan RT-PCR assays (Applied Biosystems, Foster City, CA, USA). The TaqMan MicroRNA Reverse Transcription Kit (Cat. #4366596, Applied Biosystems) was used for the preparation of cDNA. Reverse transcription reactions were performed in a volume of 15 μL, and each reaction contained 20 ng of total RNA. All TaqMan assays were run in duplicate on a Roche LC480 (Roche Diagnostic, Basel, Switzerland) using a TaqMan universal PCR master mix (Roche). Relative quantification of miRNAs expression was performed through the 2^−ΔΔCt^ method, using the average expression of reference internal control (RNU43) for normalization of miRNAs abundances across the samples. A no-template control was used as a negative control. The expression levels for each miRNA in samples were converted into a linear scale as fold-change (FC) values calculated with respect to the mean of the control samples (CTRLs). Significant miRNAs expression differences were determined using the Mann–Whitney *t*-test (*p*  ≤ 0.05), and data analysis was performed with GraphPad Prism 6.

### 2.10. Statistical Analysis of Clinical and Pathological Parameters and MiRNA Expression

All statistical analyses were performed using GraphPad Prism software Version 8. To assess the ability of miRNA expression profiles to discriminate tumor samples from normal brain tissue, Receiver Operating Characteristic (ROC) curves and the associated Area Under the Curve (AUC) were calculated. A 95% confidence interval was used.

To evaluate the predictive power of miRNAs for histological aspects (high-grade or low-grade morphology) or recurrence risk, data from the two experimental groups were compared using the non-parametric Mann-Whitney U test. A *p*-value of ≤0.05 was considered statistically significant.

## 3. Results

### 3.1. Patient Cohort Characteristics

The cohort included 14 cases (10 female and 4 male patients) histologically defined as astroblastoma, with a median age of 25 years. Eight were pediatric cases (<18 years), and the remaining cases (6 patients) were adults. All cases were supratentorial (ST). Clinical features and histology are summarized in Table 1.

### 3.2. DNA Methylation Profiling Results

DNA methylation profiling was performed on all 14 tumor samples. Eight out of 14 were classified in the methylation class “CNS high grade neuroepithelial tumour with *MN1* alteration” (HGNET-*MN1*). The remaining six cases were: one ependymoma *RELA* fusion positive, one glioblastoma mesenchymal subtype, one diffuse midline glioma *H3 K27M* mutant, one pleomorphic xanthoastrocytoma/advanced stage ganglioglioma, one CNS high grade neuroepithelial tumor with *BCOR* alteration and one glioblastoma RTK II subtype (Table 1). All cases had a calibrated score > 0.9. The median age of HGNET-*MN1* patients was 16.5 years. Although all of them were phenotypically female, one patient showed a male genotype on the methylation profile.

### 3.3. Pathological and Immunophenotypical Findings

All cases showed perivascular astroblastic pseudorosettes, variable hyalinized wall vessels and stromal sclerosis, as shown in Figure 1A–C. Immunostaining for GFAP, Olig2 and EMA was conducted, as reported in Figure 1D–F. All eight samples shared the same phenotype exhibited in Figure 1D–F. High-grade features with necrosis and mitosis were present in five cases (Figure 1C). The immunohistochemical stain for Estrogen Receptor (ER), performed later, after the results of the involved KEGG pathway, showed a faint nuclear positivity in two cases (case 3 and case 4) (Figure 1G–H).

### 3.4. Follow-Up Data

Patient follow-up details were available for 6/8 patients. One patient with histologic low-grade features presented with recurrences after 4-, 9- and 10-years post-surgery, and received Temozolomide and radiotherapy treatment. One patient with histological high-grade aspects presented several recurrences after 1-, 3-, 6-, 7-, 9- and 17-years post-surgery and after Temozolomide and radiotherapy treatment. Of the remaining four patients, one with high grade morphology went to complete remission after Bevacizumab, Temozolomide and radiation therapy. The last three patients (one with low grade morphology and two with high grade aspects) were free of disease after surgical resection.

### 3.5. Identification of Differentially Expressed miRNAs

The miRNAs expression profile was performed in tumor samples (*n* = 8) and normal brain tissues as controls (CTRLs, *n* = 2). Thirty-nine miRNAs were found to be differentially expressed (*p* < 0.05) by comparing *MN1*-altered astroblastomas versus normal controls. Among these, 22 miRNAs were upregulated (FC > +2) and 17 were downregulated (FC < −2), as reported in Table 2 and in Figure 2.

We have validated seven miRNAs out of 17 downregulated miRNAs (FC < 2.9) and five miRNAs out of 22 upregulated miRNAs (FC > 2.9) by using RT-qPCR (Appendix A). Furthermore, to assess the overall accuracy of the miRNA expression profiles in discriminating tumors from healthy samples, we employed Receiver Operating Characteristic (ROC) curves and the area under the ROC curve (AUC). Area values under the curve for upregulated and downregulated miRNAs were 0.9064 and 0.8997, respectively (*p*-value  < 0.0001 −95% confidence interval) (Figure 3).

### 3.6. Relation Between miRNA Expression and Clinico-Pathological Features

In our analysis, expression profiles of upregulated miRNAs for the 8 *MN1* astroblastomas did not show significant associations with histological grade (*p*-value: 0.3020) or recurrence risk (*p*-value: 0.4890). In contrast, downregulated miRNAs demonstrated a strong ability to differentiate both histological grades (*p*-value: 0.0079) and recurrence risk (*p*-value: <0.0001), suggesting their potential utility as biomarkers for prognostic classification (Figure 4, Table 3).

### 3.7. Epigenetic Regulation of Deregulated miRNAs by Promoter Methylation Status

To understand the underlying mechanisms of miRNAs aberrant expression, we first investigated the DNA methylation status of CpG islands in their regulatory regions, which have been identified in 32 out of 39 deregulated miRNAs. By considering family and co-clustered miRNAs, four miRNAs out of 32 miRNAs belonged to the same gene family (miR-124, miR-129, miR-181 and miR-7), and seven miRNAs were co-clustered (miR-127, miR-137, miR-181, miR-23A, miR-431, miR-494, miR-7-2).

The results showed no differences in the methylation status of the regulatory region between tumors and controls in 29 out of 32 (97%) miRNAs. The remaining three miRNAs (miR146B, miR-330, and miR-7-2) appeared downregulated in tumor samples, with a hypo-methylation status in their regulatory regions. Data are shown in Figure 5.

Among the 32 miRNAs, 14 had a coherent expression related to their promoter methylation status. More in depth, seven out of 14 hyper-expressed miRNAs (miR-10B, miR-1253, miR-146B, miR-193A, miR-210, miR-23A, miR-574) showed hypo-methylation of CpG islands (beta values < 0.2), and seven out of 14 hypo-expressed miRNAs (miR-124-2, miR-1249, miR-129-1, miR-181A1, miR-181A2, miR-181B1, miR-488) showed hyper-methylation of CpG islands (beta values > 0.8), as represented in Figure 5 and Figure 6.

### 3.8. Copy Number Variation (CNV)

To further understand the mechanisms underlying the aberrant expression of miRNAs, we investigated the DNA copy number variation (CNV) at the miRNAs mapping regions, looking for amplification, deletion, or translocations. We scanned the CNV plot of each sample in the regions where microRNAs map. None of the regions of interest had loss or gain of the specific locus that could explain the aberrant expression of deregulated miRNAs (Appendix A).

### 3.9. Functional Enrichment Analysis and Pathways Analysis

Gene Ontology (GO) enrichment analysis was conducted on the deregulated miRNAs with distinct promoter methylation status (*n* = 32) to explore further biological processes related to these miRNAs. Analyses of upregulated (*n* = 15) and downregulated (*n* = 17) miRNAs identified a list of 778 and 2.220 validated target genes, respectively. Most of the BPs of validated target genes involved are related to the regulation of gene silencing, chromatin assembly and remodelling, apoptotic signalling and regulation of the cell cycle (Appendix A). Furthermore, identification of the validated miRNA target genes was also performed. A total number of 2998 validated target genes resulted from the analysis, and are reported in Table 4.

The KEGG pathway enrichment analysis elucidated that the pathways affected by deregulated miRNAs are involved in several processes like metabolism, genetic information processing, signal transduction, cellular processes, and human diseases (Table 5).

To clarify the interaction between miRNAs and their target genes, a network enrichment analysis was also performed by MirNet, an online tool, indicating that most of the target genes are enriched in signalling cascades connected to cancer development, such as ‘Pathways in cancer’, and ‘PI3K-Akt signalling pathway’ (Figure 7).

Interestingly, an association with the MAP kinase (MAPK) signalling pathway, which has a role in the sustenance of cellular normal conduit, response to cancer therapy and activation of compensatory pathways, was found. This finding could be considered for target therapies, but additional studies are needed to confirm the evidence. Also, TGF-beta and glycosaminoglycan biosynthesis or ECM-receptor interaction pathways could explain the presence of stromal sclerosis, which is often present in these tumors. Moreover, considering the female prevalence of these tumors, the estrogenic and steroid synthesis signalling pathway was an interesting connection. In addition, we tried to evaluate the expression of the estrogenic receptor (ER) protein in our tumor samples through immunohistochemical analysis. The results showed a focal and faint expression in two out eight specimens (Figure 1G–H). Among the pathways we investigated, MAPK signaling, TGF-beta signaling, glycosaminoglycan biosynthesis, and estrogenic and steroid synthesis were associated with various miRNAs. Notably, the miR-30 miRNA family was commonly expressed across all these pathways.

## 4. Discussion

Astroblastoma is a rare glial tumor characterized by structural rearrangements of the MN1 gene at chromosome 22q12.1 [7]. Histological findings range from papillary to solid aspects, but the main characteristic is the presence of perivascular pseudorosettes, vascular hyalinization, and lack of fibrillary background [44]. The glial origin is sustained by the positivity for GFAP and OLIG2, and the common positivity for EMA may suggest a transitional glial—ependymal pattern. Cellular pleomorphism, vascular proliferation, necrosis, and high proliferation rate can be present and associated with worse prognoses [45]. However, aside from surgical resection, no additional prognostic factors have been identified [12]. Because of their variable biological behavior [12,44,45,46], in the 2021 WHO classification for CNS tumors, no grade is given [47].

Since similar histological features may be present in other tumors with different molecular alterations and prognosis, the detection of *MN1* rearrangement represents a fundamental molecular approach for the diagnosis of astroblastoma.

Our study comprises eight cases defined by DNA methylation profile, with an optimal score in the methylation class “CNS high grade neuroepithelial tumor with *MN1* alteration”. Histological characteristics comprise the perivascular and papillary pattern, the presence of astroblastic pseudorosettes, and vessels hyalinosis. Some cases show stromal sclerosis. Histological aspects of malignancy are present in five samples, whereas three cases reveal low grade histological aspects.

Genome-wide studies have demonstrated a close relationship between miRNAs deregulations and human cancers [48]. The MiRNA expression signature is associated with tumor type, tumor grade and/or clinical outcomes. Here, we analyzed miRNA expression in tumor tissues of the eight *MN1* alterate astroblastomas and two normal controls. A total of 39 miRNAs are deregulated compared to normal brain tissues: 22 are overexpressed and 17 are downregulated. We also analyzed the relationship between these deregulated microRNAs and clinicopathological features, revealing an intriguing pattern. Upregulated miRNAs showed no significant correlation with histological grade or recurrence risk. In contrast, downregulated miRNAs demonstrated a strong association with an increased risk of recurrence.

Recent studies on gliomas have supported these findings. For instance, the let-7 family of miRNAs, known for their tumor-suppressing functions, has been observed to be downregulated in glioma recurrence, regardless of WHO tumor grade contributing to enhanced cell proliferation and resistance to apoptosis [49]. This finding underscores the potential of downregulated miRNAs as prognostic biomarkers. Mechanistically, it raises the possibility that their decreased expression might impair tumor-suppressive pathways, such as those regulating cell cycle control, apoptosis, or immune response. Specific miRNAs, such as miR-30 and let-7, have highlighted similar roles in other malignancies, suggesting a broader relevance.

Clinically, these results could pave the way for miRNA-based assays to identify high-risk glioma patients more accurately, enabling personalized treatment strategies. For instance, integrating miRNA profiling into standard histopathological evaluation may improve predictions of recurrence risk. Furthermore, therapeutic strategies aimed at restoring the function of downregulated miRNAs could offer a promising direction. Also, this miRNA signature holds potential diagnostic and classification implications. Currently, the WHO classification of central nervous system (CNS) tumors increasingly integrates molecular and genetic alterations alongside histopathological criteria. Our results suggest that the unique miRNA expression profiles associated with *MN1*-altered astroblastoma could serve as an additional molecular marker for more precise tumor categorization. Specifically, these miRNA signatures could complement existing diagnostic markers, aiding in distinguishing *MN1*-altered astroblastoma from other gliomas and morphologically similar entities. By expanding the molecular criteria for astroblastoma, the WHO classification could better account for the unique biological and clinical characteristics, ultimately improving patient diagnosis and management.

The aberrant miRNAs expression in cancer can occur via various mechanisms [23], which include epigenetic [50] and genomic abnormalities [20,25,26], transcriptional deregulation [31], and altered miRNAs processing [51,52,53].

Related to epigenetic changes and the consequential modulation of miRNAs expression, we deepened the methylation status of CpG islands on miRNAs promoters at the core level and at different distances from the TSS. The analysis of the known promoter regions of 32 deregulated miRNAs (consisting of 15 up- and 17 down-regulated) reveals that 14 are epigenetically modulated by their promoter. Therefore, 7 miRNAs are overexpressed while their promoters are hypomethylated, and 7 miRNAs are downregulated by their hypermethylated promoters. The expression of the remaining 18 miRNAs is not dependent on the methylation status of their promoters. Furthermore, we found the same methylation trend of the promoter regions between tumor samples and normal brain tissue on 29 out 32 miRNAs. On the other hand, three of the downregulated miRNAs (miR146B, miR-330 and miR-7-2) in tumor tissues have an opposite tendency to high methylation levels of their promoter compared to a normal brain. Next, we explored the CNV of tumor samples to evaluate if the expression of the 18 miRNAs not epigenetically regulated are dependent on genomic abnormal ties. The analysis of the chromosomic region where miRNAs are located did not highlight significant deletions or gains to explain the aberrant expression of these miRNAs.

MiRNAs alteration often leads to target gene deregulation and deregulated signalling pathways with roles in tumor development and progression [54].

We identified validated gene targets for each deregulated miRNAs and conducted GO enrichment for their BPs. The 15 upregulated miRNAs control 778 genes, and the 17 downregulated miRNAs operate on 2220 genes. The identification of key biological pathways influenced by miRNA regulation provides insight into tumor biology and potential therapeutic vulnerabilities. For example, miRNAs implicated in pathways associated with cell proliferation, apoptosis, and differentiation may reflect tumor-specific mechanisms that could be targeted in future treatment strategies.

Moreover, the KEGG pathways of deregulated miRNAs curiously highlight an association with TGF-beta and glycosaminoglycan biosynthesis, as well as the ECM-receptor interaction pathway which could be related to stromal and vessels hyalinosis. This association has been described in the literature for miR-30 family members, where authors described a relationship between miRNAs and liver fibrosis by regulating TGF-beta signaling in hepatic stellate cells [55,56]. Another interesting pathway implicated in deregulated miRNAs is the estrogenic and steroid synthesis signaling pathway [57], considering the female prevalence of these tumors. Howard and Yang described miRNAs association with ER-positive and ER-negative breast cancer subtypes [58]. Additionally, the involvement of the MAPK signaling pathway could be important for investigating potential target therapies. Related to this hypothesis, some authors demonstrated that the overexpression of miR-30E inhibits the adhesion, migration and cell cycle progression in prostate cancer cells suppressing the activation of the MAPK signaling pathway [59].

MN1-altered astroblastoma tumors have a miRNA expression signature that identifies a prognostic correlation, specific BPs, and molecular pathways in these tumors, revealing critical regulatory functions at multiple levels. We acknowledge that the results presented in this study must be interpreted with caution due to the limited sample size, which represents an inherent limitation in rare tumor studies like astroblastoma. A small cohort can affect the statistical power and generalizability of the findings, and while the bioinformatics analyses provide valuable insights, they require further validation to ensure robustness and biological significance.

To address these limitations, we plan to conduct additional wet lab experiments in future studies with techniques that will allow us to confirm miRNA expression levels and their downstream effects on target mRNAs and proteins, providing direct experimental evidence to support bioinformatics predictions.

Moreover, we recognize the importance of investigating the transcriptional regulation of these miRNAs to better understand their functional roles in astroblastoma pathogenesis. Transcriptional regulation studies could involve promoter analysis, chromatin immunoprecipitation (ChIP), or RNA sequencing to elucidate the upstream factors influencing miRNA expression. Such studies would deepen our understanding of the molecular mechanisms underlying this rare tumor and potentially uncover novel therapeutic targets.

We emphasize that increasing the sample size and integrating wet lab validations are essential next steps to improve the reliability and translational impact of this research. These future efforts will focus on bridging the gap between bioinformatics predictions and biological relevance, leading to a more comprehensive understanding of miRNA-mediated regulation in astroblastoma.

## Figures and Tables

**Figure 1 biomedicines-13-00112-f001:**
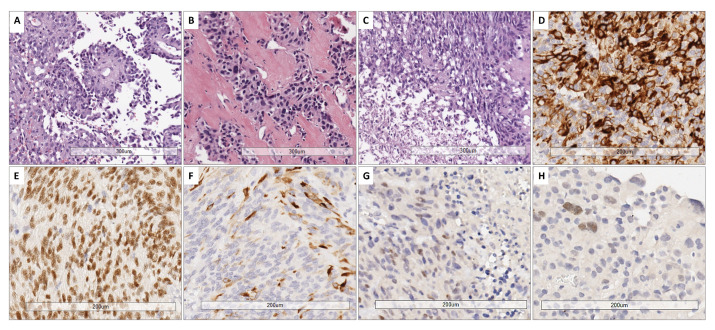
Representative histological features. (**A**) Papillary-like arrangement of astroblastic pseudorosettes with polygonal perivascular cells. (**B**) Area of low cellularity with dense vascular and stromal collagenous sclerosis. (**C**) High-grade astroblastoma showing anaplasia and foci of necrosis (below on the left part) (H&E. 10× magnification). (**D**) Cytoplasmic GFAP expression. (**E**) Nuclear expression of Olig2. (**F**) Variable positivity for EMA (20× magnification). (**G**,**H**) Estrogen receptor (ER) immunostain exhibited a faint nuclear staining in 2 cases (G: case 3; H: case 4) (20× magnification).

**Figure 2 biomedicines-13-00112-f002:**
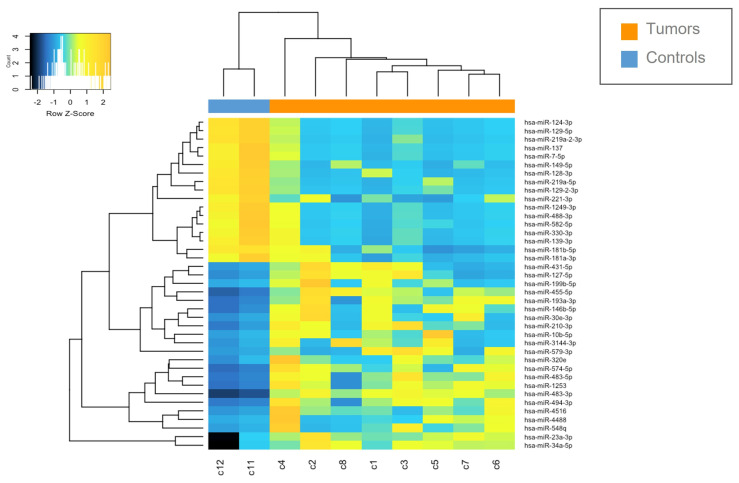
Heatmap of deregulated miRNAs. Hierarchical clustering (Euclidian distance, complete linkage) of the eight MN1-altered astroblastoma samples (C1–C8) and two CTRLs (C11–C12) based on the expression profile of the 39 deregulated miRNAs. The heatmap shows miRNAs with high expression in yellow and miRNAs with low expression in blue. The heatmap was generated by an online tool (http://www.heatmapper.ca/expression/ (accessed on 1 January 2021)).

**Figure 3 biomedicines-13-00112-f003:**
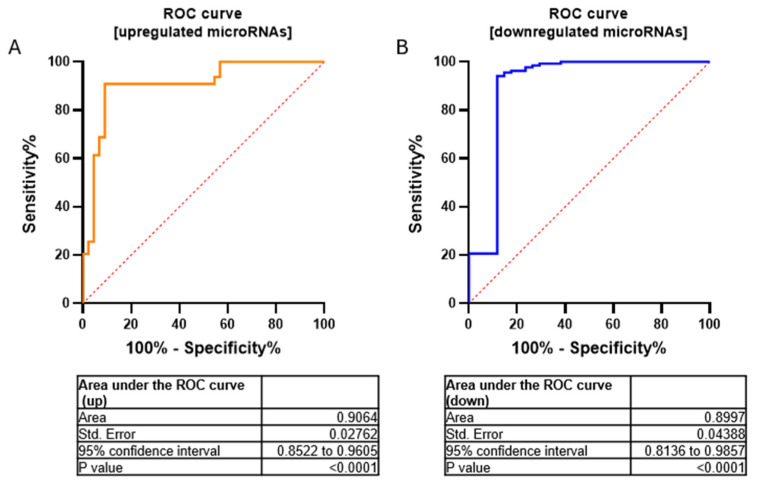
ROC curves for upregulated (**A**) and downregulated (**B**) microRNAs in a cohort of MN1-altered astroblastoma samples and the normal control. Area values under the curve were AUC  =  0.9064 and 0.8997, respectively; *p*-value  < 0.0001 using 95% confidence interval.

**Figure 4 biomedicines-13-00112-f004:**
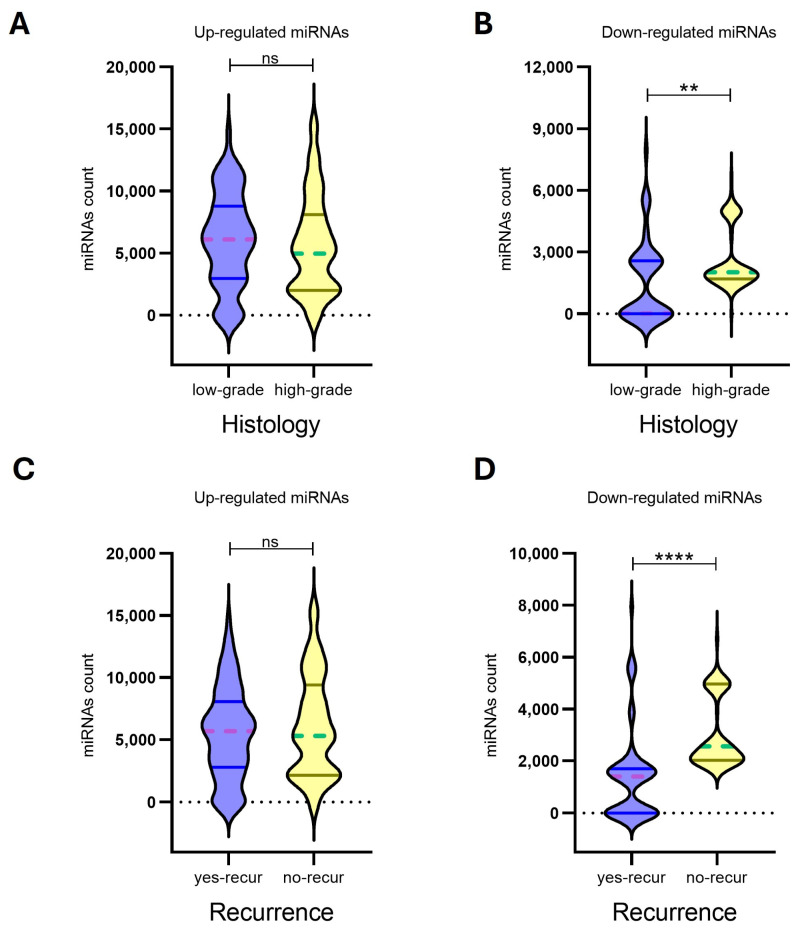
The expression levels of up-regulated miRNAs (*n* = 22) (**A**,**C**) and down-regulated miRNAs (*n* = 17) (**B**,**D**) in eight MN-1 altered astroblastoma patients. MiRNA expression was evaluated by Nanostring nCounter, with counts reported for each miRNA. Patients with *MN-1* altered astroblastoma were categorized based on their histology into (**A**) low-grade tumors, (**B**) high-grade tumors and (**C**,**D**) recurrence. A *p*-value of ≤0.05 was considered statistically significant, (**) *p* < 0.005, (****) *p* < 0.0001, while ns indicates not statistically significant. In the violin plots, whole lines represent quartiles, and dotted lines indicate median values.

**Figure 5 biomedicines-13-00112-f005:**
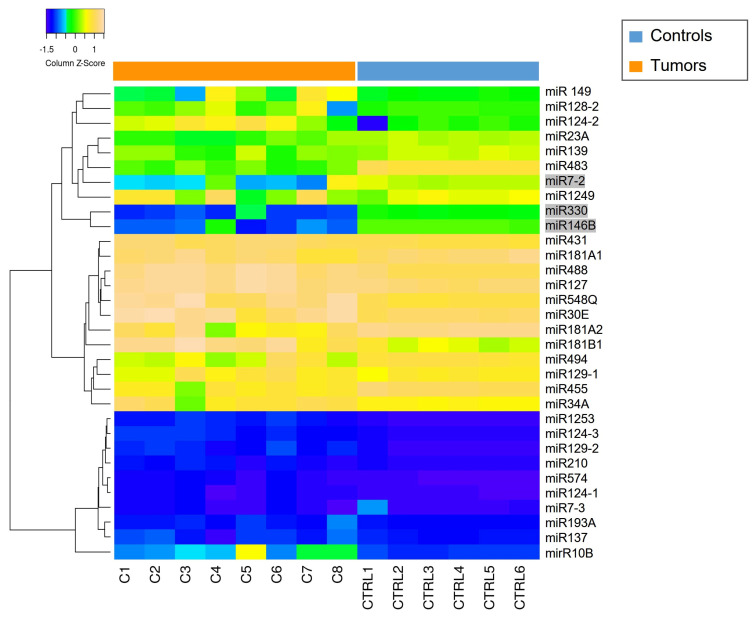
Methylation status of miRNAs promoter (total = 32) in tumor samples (C1–C8) and CTRLs (CTRL 1–CTRL6). The heatmap shows high levels of methylation in yellow and low levels of methylation in blue. MiR-7-2, miR-330 and miR-146B (highlighted with a gray background) exhibit promoter methylation status that are not consistent with those observed in the controls. The heatmap was generated by an online tool (http://www.heatmapper.ca/expression/ (accessed on 1 January 2021)).

**Figure 6 biomedicines-13-00112-f006:**
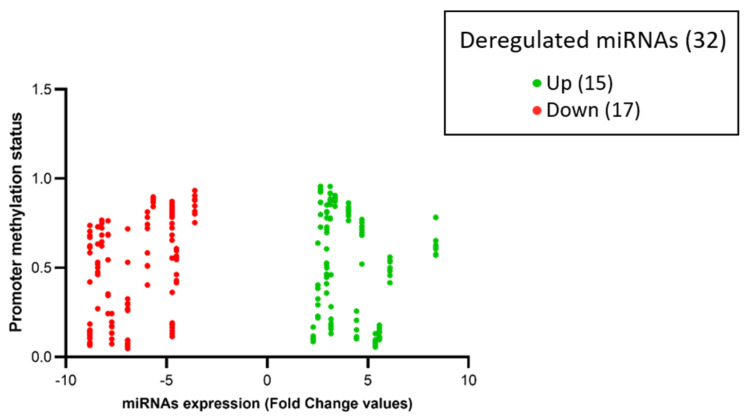
The volcano-plot illustrates the relation between promoter methylation status and the expression of specific miRNAs. The vertical axis (*y*-axis) corresponds to the methylation levels (beta-value from 0 to 1) of the promoters of deregulated miRNAs. The horizontal axis (*x*-axis) represents deregulated miRNAs: green dots represent upregulated miRNAs and red dots represent downregulated miRNAs expressed in fold change values (generated using Prism GraphPad Software Version 8.0.2.).

**Figure 7 biomedicines-13-00112-f007:**
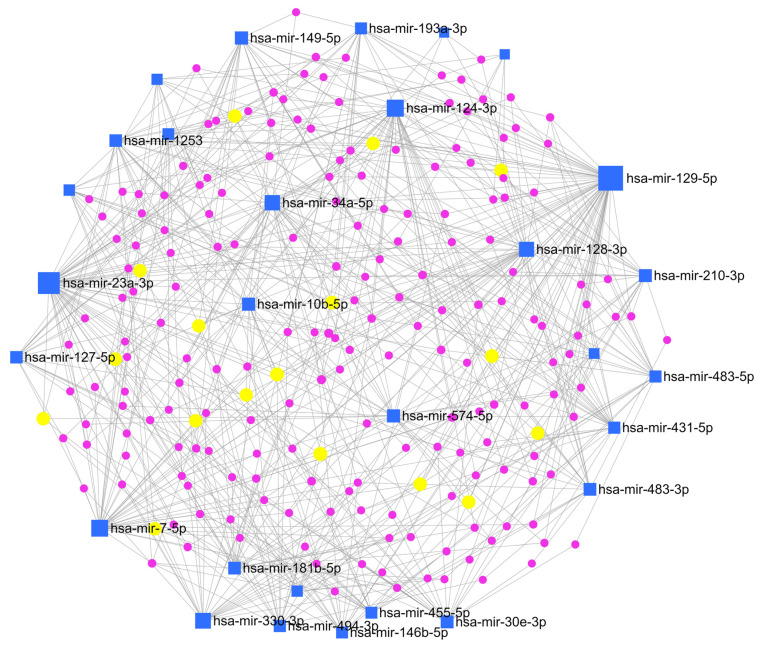
Network enrichment analysis. miRNAs-gene network was constructed based on miRNAs signature by using miRNet (online tool https://www.mirnet.ca (accessed on 1 January 2021)). Blue squares indicate miRNAs, violet dots indicate genes, and yellow dots indicate genes involved in the KEGG pathway of interest for astroblastoma (see text below).

**Table 1 biomedicines-13-00112-t001:** Clinical patient characteristics, tumor histology, methylation profiling with calibrated sore.

Samples	Sex	Age at Diagnosis (Years)	Localization	Histology	Methylation Profile	Calibrated Score (v11b4)
1	F	17	Left parietal lobe	AB	CNS HGNET-*MN1*	0.98
2	F	17	Left parietal lobe	AB	CNS HGNET-*MN1*	0.99
3	F	19	Right frontal lobe	AB	CNS HGNET-*MN1*	0.99
4	F	5	Right parietal lobe	HG AB	CNS HGNET-*MN1*	0.99
5	F	6	Right temporal lobe	HG AB	CNS HGNET-*MN1*	0.99
6	F	10	Left temporal lobe	HG AB	CNS HGNET-*MN1*	0.99
7	F	11	Right parietal lobe	HG AB	CNS HGNET-*MN1*	0.99
8	F	47	Right parietal lobe	HG AB	CNS HGNET-*MN1*	0.99
9	M	16	Left frontal lobe	AB	*RELA* fusion-positive EPN	0.99
10	M	57	Right temporal lobe	AB	GBM MES	0.99
11	M	16	Right thalamus	HG AB	*K27M* DMG	0.99
12	F	31	Right temporal lobe	HG AB	PXA	0.99
13	F	40	Left frontal lobe	HG AB	CNS HGNET-*BCOR*	0.99
14	M	60	Right insular lobe	HG AB	GBM RTK II	0.99

Abbreviations: Astroblastoma (AB), High grade astroblastoma (HG AB), CNS high-grade neuroepithelial tumor with *MN1* alteration (CNS HGNET-*MN1*), Ependymoma *RELA* fusion positive (RELA fusion-positive EPN), Glioblastoma mesenchymal subtype (GBM MES), Diffuse midline glioma *H3K27M* mutant (*K27M* DMG), Pleomorphic astrocytoma (PXA), CNS high-grade neuroepithelial tumor with *BCOR* alteration (CNS HGNET-*BCOR*), Glioblastoma RTK II subtype (GBM RTK II).

**Table 2 biomedicines-13-00112-t002:** List of differentially expressed miRNAs in *MN-1* altered astroblastoma patients compared to normal controls.

Upregulated (*n* = 22)	Downregulated (*n* = 17)
miR-10b-5p	miR-124-3p
miR-1253	miR-1249-3p
miR-127-5p	miR-128-3p
miR-146b-5p	miR-129-2-3p
miR-193a-3p	miR-129-5p
miR-199b-5p	miR-137
miR-210-3p	miR-139-3p
miR-23a-3p	miR-149-5p
miR-30e-3p	miR-181a-3p
miR-3144-3p	miR-181b-5p
miR-320e	miR-219a-2-3p
miR-34a-5p	miR-219a-5p
miR-431-5p	miR-221-3p
miR-4488	miR-330-3p
miR-4516	miR-488-3p
miR-455-5p	miR-582-5p
miR-483-3p	miR-7-5p
miR-483-5p	
miR-494-3p	
miR-548q	
miR-574-5p	
miR-574-3p	

**Table 3 biomedicines-13-00112-t003:** Correlation between differentially expressed miRNAs and clinicopathological parameters in eight *MN-1* altered astroblastoma patients. Analyses were performed among experimental groups by using a nonparametric Mann–Whitney U *t*-test. The median of miRNAs counts expression were reported for up-regulated and down-regulated miRNAs. A *p*-value of ≤0.05 was considered statistically significant.

miRNAs Expression				
Clinical Parameters	Up-Regulated		Down-Regulated	
	Median of miRNAs Counts Expression; *p*-Value			
Histology (*n* = 8)				
Low-grade (*n* = 3)	6113	*p* = 0.3020	1700	*p* = 0.0079
High-grade (*n* = 5)	4962		2011	
Recurrence (*n* = 8)				
Yes (*n* = 4)	5694	*p* = 0.489	1401	*p* < 0.0001
No (*n* = 4)	5302		2563	

**Table 4 biomedicines-13-00112-t004:** Deregulated miRNAs with known deregulated methylation promoter status and the total number of their validated target genes.

Validated Target Genes of Deregulated miRNAs	
miRNAs	Target Genes
Upregulated (*n* = 15)	778
Downregulated (*n* = 17)	2220

**Table 5 biomedicines-13-00112-t005:** KEGG Pathway analysis: Union of involved pathways of deregulated miRNAs.

KEGG Pathway	*p*-Value	Genes Count
Adherens junction	6.55 × 10^−6^	65
Fatty acid metabolism	5.96 × 10^−2^	33
Proteoglycans in cancer	8.88	137
Cell cycle	24.7	95
Protein processing in endoplasmic reticulum	5.41 × 10^2^	120
Pathways in cancer	2.12 × 10^3^	251
Viral carcinogenesis	1.79 × 10^4^	134
Hippo signaling pathway	1.89 × 10^4^	97
Colorectal cancer	3.97 × 10^4^	50
Fatty acid biosynthesis	5.41 × 10^4^	8
Chronic myeloid leukemia	5.41 × 10^4^	57
Endocytosis	1.72 × 10^5^	136
Pancreatic cancer	2.54 × 10^4^	52
Hepatitis B	3.32 × 10^5^	95
p53 signaling pathway	9.89 × 10^5^	54
Bacterial invasion of epithelial cells	4.00 × 10^6^	55
Renal cell carcinoma	4.00 × 10^6^	49
Prion diseases	4.22 × 10^6^	19
Prostate cancer	4.22 × 10^6^	65
Regulation of actin cytoskeleton	8.55 × 10^6^	134
Fatty acid elongation	1.12 × 10^−4^	15
Thyroid hormone signaling pathway	3.09 × 10^−4^	79
Non-small cell lung cancer	3.83 × 10^−4^	42
Ubiquitin mediated proteolysis	4.17 × 10^−4^	96
HTLV-I infection	4.34 × 10^−4^	165
Small cell lung cancer	5.37 × 10^−4^	61
Axon guidance	5.88 × 10^−4^	75
Glioma	7.42 × 10^−4^	44
DNA replication	8.42 × 10^−4^	27
FoxO signaling pathway	8.42 × 10^−4^	87
Transcriptional misregulation in cancer	8.57 × 10^4^	116
Oocyte meiosis	9.62 × 10^−4^	74
Focal adhesion	9.64 × 10^−4^	131
Arrhythmogenic right ventricular cardiomyopathy (ARVC)	1.52 × 10^−3^	42
Shigellosis	1.59 × 10^−3^	44
ECM-receptor interaction	1.61 × 10^−3^	44
Insulin signaling pathway	1.67 × 10^−3^	92
Neurotrophin signaling pathway	2.22 × 10^−3^	79
ErbB signaling pathway	2.52 ×10^−3^	58
Endometrial cancer	2.52 × 10^−3^	38
Acute myeloid leukemia	3.24 × 10^−3^	41
Biotin metabolism	3.33 × 10^−3^	3
Progesterone-mediated oocyte maturation	3.46 × 10^−3^	60
Thyroid cancer	3.50 × 10^−3^	23
HIF-1 signaling pathway	4.03 × 10^−3^	69
MAPK signaling pathway	4.19 × 10^−3^	150
mRNA surveillance pathway	4.40 × 10^−3^	62
Signaling pathways regulating pluripotency of stem cells	5.19 × 10^−3^	88
VEGF signaling pathway	5.51 × 10^−3^	43
Sphingolipid signaling pathway	5.51 × 10^−3^	76
Gap junction	6.05 × 10^−3^	57
Bladder cancer	6.18 × 10^−3^	29
Salmonella infection	1.07 × 10^−2^	56
Fc gamma R-mediated phagocytosis	1.30 × 10^−2^	59
TGF-beta signaling pathway	1.55 × 10^−2^	50
Base excision repair	1.56 × 10^−2^	20
Lysine degradation	1.77 × 10^−2^	30
Choline metabolism in cancer	1.88 × 10^−2^	64
Central carbon metabolism in cancer	2.20 × 10^−2^	42
TNF signaling pathway	2.72 × 10^−2^	65
Sulfur metabolism	2.89 × 10^−2^	7
Biosynthesis of unsaturated fatty acids	2.89 × 10^−2^	13
Apoptosis	2.89 × 10^−2^	56
Pathogenic Escherichia coli infection	3.30 × 10^−2^	37
RNA transport	3.38 × 10^−2^	100
mTOR signaling pathway	3.38 × 10^−2^	40
Long-term depression	3.75 × 10^−2^	36
Phosphatidylinositol signaling system	3.86 × 10^−2^	51
Fatty acid degradation	4.43 × 10^−2^	22
Estrogen signaling pathway	4.43 × 10^−2^	61
Spliceosome	4.54 × 10^−2^	78
Melanoma	4.54 × 10^−2^	43
Lysosome	4.74 × 10^−2^	73
Glycosaminoglycan biosynthesis-chondroitin sulfate/dermatan sulfate	4.89 × 10^−2^	12

## Data Availability

Dataset available on request from the corresponding author.

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
