# Peer review of "MicroRNAs Expression Profile in MN1-Altered Astroblastoma"

_biomedicines, 2025, doi:10.3390/biomedicines13010112_

Round 1
Reviewer 1 Report
Comments and Suggestions for Authors
1. Introduction section, the authors did not elaborate on the importance of this study. The introduction about the research status and significance of microRNAs in astroblastoma is insufficient. It is suggested to further explain the role of microRNAs as gene expression regulators in tumor development, as well as some important findings of microRNA expression profile studies in other gliomas.
2. Section 2.3, the process of DNA methylation and miRNA expression profile analysis should be more detailed, such as specific quality control criteria and batch effect correction information.
3. Most of the research in this paper focuses on bioinformatics analysis, lacking wet lab validation of miRNA target genes, such as qPCR and WB experiments to confirm the regulatory relationship of key miRNA-mRNA. It is suggested that the authors expand the relevant discussion.
4. The analysis of prognosis or clinical relevance could be further explored, such as attempting to construct a prognostic classification model based on miRNA expression profiles and exploring the correlation between miRNA markers and survival.
5. In Section 3.5, for miRNA expression profile analysis, the authors used a p-value threshold of 0.05 to screen for differentially expressed miRNAs. However, considering the large number of miRNAs tested, it is suggested to consider using the q-value after FDR correction as the screening criterion to reduce false positives.
6. Line 302, the authors mentioned Table 4, but this table is missing and needs to be added.
7. Lines 319-324, presenting the ER staining results here seems abrupt. Consider moving it to the previous pathological results section and discussing it together with other markers.
8. Lines 337-338, the authors mentioned that in the 2021 WHO CNS classification criteria, astroblastoma has not yet been defined as a distinct molecular entity. It is suggested that the authors could expand the discussion here and explain what implications the new findings of this study may have for the improvement of the WHO classification criteria.
9. The last paragraph of the discussion (Lines 390-393), it is suggested that the authors should also evaluate the limitations of the study and propose prospects for future research directions.
Author Response
1.Introduction section, the authors did not elaborate on the importance of this study. The introduction about the research status and significance of microRNAs in astroblastoma is insufficient. It is suggested to further explain the role of microRNAs as gene expression regulators in tumor development, as well as some important findings of microRNA expression profile studies in other gliomas.
We appreciate the reviewer’s feedback and acknowledge the need to enhance the Introduction section by elaborating on the importance of our study. Specifically, we have addressed the following points:
The Role of MicroRNAs as Gene Expression Regulators in Tumor Development
We have added a detailed explanation of the critical role of microRNAs (miRNAs) in regulating gene expression, emphasizing their dual functions as oncogenes or tumor suppressors genes. This discussion highlights their ability to modulate key processes such as cell proliferation, apoptosis, angiogenesis, and metastasis, which are essential in tumor development and progression.
MicroRNA Expression Profiles in Other Gliomas
We have incorporated an overview of significant findings from previous studies on miRNA expression profiles in gliomas. This includes examples of miRNA expression in glioblastomas, miRNA signature in diffuse intrinsic pontine gliomas (DIPGs) as well as low grade gliomas
We have revised the Introduction section to include these additional details (lines: 84-95 and 103-115).
- Section 2.3, the process of DNA methylation and miRNA expression profile analysis should be more detailed, such as specific quality control criteria and batch effect correction information.
We thank the reviewer for the observation, we have provided details in the main text (Paragraph 2.3, line 171-175; Paragraph 2.7, lines 222-227). For miRNA expression profile, we preferred to conduct a single Nanostring run (a single run can accommodate 12 samples) from individual patients to avoid batch effects when examining miRNA expression over time.
- Most of the research in this paper focuses on bioinformatics analysis, lacking wet lab validation of miRNA target genes, such as qPCR and WB experiments to confirm the regulatory relationship of key miRNA-mRNA. It is suggested that the authors expand the relevant discussion.
We sincerely appreciate the reviewer’s comment. We acknowledge the importance of validating the findings obtained through bioinformatics analysis, and we agree that experimental confirmation is crucial to strengthening the conclusions of our study. We validated 7 out of 17 downregulated miRNAs and 5 out of 22 upregulated miRNAs using RT-qPCR, as reported in Supplementary Figure 1. Primarily, we focused this study on bioinformatic analysis and identification of potential miRNA-mRNA interactions. We plan to pursue further validation in future experiments, to confirm the regulatory relationships of the key miRNAs identified in this study. Additionally, we agree that investigating the transcriptional regulation of these miRNAs will provide valuable insights into their functional roles in the context of astroblastoma. We have expanded the relevant discussion in the revised manuscript to address the need for experimental validation and to highlight our future directions for experimental validation (Paragraph 4).
- The analysis of prognosis or clinical relevance could be further explored, such as attempting to construct a prognostic classification model based on miRNA expression profiles and exploring the correlation between miRNA markers and survival.
We thank the reviewer for the observation.
To test the specificity and sensitivity of the miRNA expression profile in discriminating tumors from healthy samples we employed Receiver Operating Characteristic (ROC) curves and the area under the ROC curve (AUC). To further explore the clinical relevance and prognostic potential of the 39 deregulated miRNAs, we correlated the up- or downregulated miRNAs with histological features (high- grade VS low-grade morphologic aspects) and with recurrence risk.
These findings indicate the potential of downregulated miRNAs to serve as valuable indicators for clinical outcomes, offering a basis for developing predictive models for survival and recurrence risk. Further efforts could focus on constructing a comprehensive prognostic classification model based on miRNA expression profiles and validating the correlation between miRNA markers and overall survival in larger, independent cohorts.
We introduced this information adding the paragraph 2.10 in the “Materials and Methods” section; the paragraphs 3.5 and 3.6 in the “Results” section, and the relative Figure 3, Figure 4 and Table 3. We also expanded the results in the discussion.
- In Section 3.5, for miRNA expression profile analysis, the authors used a p-value threshold of 0.05 to screen for differentially expressed miRNAs. However, considering the large number of miRNAs tested, it is suggested to consider using the q-value after FDR correction as the screening criterion to reduce false positives.
We thank the reviewer for the observation. We agree that the correction of p-values ​​in the case of multiple tests reduces the risk of false positives, however for this comparison the sample size is very low (n = 10, 2 vs 8) and the correction of p-values ​​could exclude a large proportion of true positives. Correcting the p-values ​​using the FDR method no miRNA shows a p-value below the significance threshold of 0.05, however, the validations performed using RT-qPCR on 12 of the 39 DE miRNAs support the validity of the results and highlight the risk of excluding potentially relevant results due to a too stringent statistical filter in the case of a very low sample size.
- Line 302, the authors mentioned Table 4, but this table is missing and needs to be added.
We apologize for the oversight regarding Table 4. This table was inadvertently omitted during the submission process. We have now included Table 5 (previously table 4) in the revised manuscript, ensuring that all referenced tables are present and properly formatted.
- Lines 319-324, presenting the ER staining results here seems abrupt. Consider moving it to the previous pathological results section and discussing it together with other markers.
We appreciate the reviewer’s suggestion regarding the placement of the ER staining results in Figure 1. This reorganization provides a more logical presentation of the immunohistologic data, however, we preferred to leave the expression of estrogen in the discussion section as it does not have a diagnostic value (as for the other immunohistochemical analyses performed). The staining was conducted after the results of the KEGG pathway analyses, suggesting a potential involvement of estrogen-related signaling. This step was taken to validate and strengthen the bioinformatics findings. We hope these clarify the flow of the manuscript, and we thank the reviewer for pointing out this opportunity for improvement.
- Lines 337-338, the authors mentioned that in the 2021 WHO CNS classification criteria, astroblastoma has not yet been defined as a distinct molecular entity. It is suggested that the authors could expand the discussion here and explain what implications the new findings of this study may have for the improvement of the WHO classification criteria.
We appreciate the reviewer’s suggestion to elaborate on the implications of our findings for improving the WHO classification criteria for astroblastoma. We have expanded the discussion to address this point and highlight how our study contributes to refining the diagnostic and molecular understanding of astroblastoma (Paragraph 4)
- The last paragraph of the discussion (Lines 390-393), it is suggested that the authors should also evaluate the limitations of the study and propose prospects for future research directions.
We appreciate the reviewer’s valuable suggestion to evaluate the limitations of our study and propose future research directions. In the revised manuscript, we have added, in the discussion, the limitations of this study. Specifically, we acknowledge the following:
- The relatively small sample size, limits the statistical power and generalizability of our findings.
- The reliance on bioinformatics analyses without experimental validation, which necessitates further wet lab experiments to confirm the predicted miRNA-mRNA interactions.
To address these limitations, we propose several directions for future research:
- Expanding the cohort size to improve statistical robustness and allow subgroup analyses.
- Conducting qPCR, Western Blot, and functional assays to validate the regulatory relationships of key miRNAs identified in this study.
- Investigating the transcriptional and epigenetic regulation of these miRNAs to better understand their upstream control mechanisms.
- Exploring the potential clinical applications of these miRNA signatures, such as their use as biomarkers for diagnosis, prognosis, and therapeutic targeting.
We have integrated these points into the revised discussion to provide a more comprehensive perspective on the study’s scope and its implications for future research (Paragraph 4).
Reviewer 2 Report
Comments and Suggestions for Authors
The article titled “MicroRNAs Expression Profile in Astroblastoma, MN1-Altered” discussed the miRNAs expression signature in MN1-altered astroblastomas. Despite the small sample size, this article still has some novelty, but some concerns should be addressed:
1. Ethic approval number of human patient samples should be provided in the methods section.
2. In table 1, what methods and standard did the authors use to distinguish the methylation profile?
3. Are the 6 images in Figure 1 from one patient? Do all 8 CNS HGNET-MN1 samples have the same phenotype showed in Figure 1?
4. In table 3, what data set did the authors use when they analyzed the data to get 778/2220 target genes?
5. Scale bar should be added to Figure 1 and Figure 6.
Author Response
- Ethic approval number of human patient samples should be provided in the methods section.
We thank the reviewer for highlighting the need to include the ethics approval information.
The ethics approval number for the use of human patient samples has been added to the Methods section in the revised manuscript (Paragraph 2.1). Specifically, the study received the full approval from the local ethics committee of Policlinico Umberto I of Rome (approval code: Prot 0102463; approval date: 24/06/2020) and the subjects or their parents gave written informed signed to their informed consent to the use of their biological material and data for research purposes. All procedures were conducted in accordance with ethical standards and guidelines. We appreciate the reviewer’s attention to this important detail.
- In table 1, what methods and standard did the authors use to distinguish the methylation profile?
We thank the reviewer for the question.
To distinguish the methylation profile, we utilized the Heidelberg Brain Tumour Classifier (version 11b4). This classifier compares the generated methylation data against a reference dataset encompassing 91 distinct brain tumour entities. The analysis assigns a subgroup score, indicating the likelihood of the tumor belonging to a specific methylation class. The classifier's methodology relies on a robust machine-learning framework trained on well-characterized methylation profiles from a large cohort of brain tumor samples, ensuring high specificity and sensitivity for subgroup assignment. A column indicating the calibrated score has been added to Table 1.
- Are the 6 images in Figure 1 from one patient? Do all 8 CNS HGNET-MN1 samples have the same phenotype showed in Figure 1?
We thank the reviewer for their question.
The six images in Figure 1 do not all belong to the same patient. These images were selected as representative examples of the histological and immunohistochemical findings observed across all eight CNS HGNET-MN1 samples.
Additionally, we confirm that all eight samples share the same phenotype shown in Figure 1, supporting the uniformity of the features across the cohort. We have clarified this in the revised figure legend to avoid any ambiguity.
- In table 3, what data set did the authors use when they analyzed the data to get 778/2220 target genes?
Thank you for your observation. We have clarified the dataset of miRNAs analyzed for each GO enrichment analysis in the main text. As mentioned in the paragraph 3.8, “Gene Ontology (GO) enrichment analysis was conducted on the deregulated miRNAs with distinct promoter methylation status (n=32) to explore further the biological processes related to these miRNAs.” Thank you for your feedback; we have revised the text accordingly.
- Scale bar should be added to Figure 1 and Figure 6.
We appreciate the reviewer’s suggestion to add scale bars to the histologic figures to enhance the clarity and interpretability of the images. The figure legends have also been updated to indicate the magnification and scale bar measurements for each panel. Thank you for highlighting this important improvement.
Round 2
Reviewer 1 Report
Comments and Suggestions for Authors
The author clearly answered all the questions previously posed, and the manuscript could be accepted in this form.
Reviewer 2 Report
Comments and Suggestions for Authors
This manuscript has been sufficiently improved to warrant publication in Biomedicines.